# Functional Roles of the Conserved Amino Acid Sequence Motif C, the Antiporter Motif, in Membrane Transporters of the Major Facilitator Superfamily

**DOI:** 10.3390/biology12101336

**Published:** 2023-10-16

**Authors:** Manuel F. Varela, Anely Ortiz-Alegria, Manjusha Lekshmi, Jerusha Stephen, Sanath Kumar

**Affiliations:** 1Department of Biology, Eastern New Mexico University, Portales, NM 88130, USA; anely.ortizalegria@enmu.edu; 2ICAR-Central Institute of Fisheries Education (CIFE), Mumbai 400061, India; manjusha@cife.edu.in (M.L.); jerusha.phtpa702@cife.edu.in (J.S.); sanathkumar@cife.edu.in (S.K.)

**Keywords:** antiporter motif, motif C, major facilitator superfamily, transporter, antimicrobial resistance, multidrug efflux, bacteria, cancer, drug resistance

## Abstract

**Simple Summary:**

A phospholipid membrane covers all living cells, forming an impenetrable barrier circumvented by solute transporters in the cell membrane. These proteins comprise energy-requiring systems, called active transporters, and those not requiring energy, called passive transporters. The major facilitator superfamily harbors thousands of transport proteins found in all living organisms, from bacteria to humans. Alignments of multiple amino acid sequences uncovered highly conserved sequence motifs are known to play important functional roles. One of these conserved sequences, the antiporter sequence motif or motif C, participates in the molecular mechanism of antimicrobial efflux in cancer cells and bacterial pathogens. The biological implications of the antiporter motif’s functional roles and usefulness are considered here.

**Abstract:**

The biological membrane surrounding all living cells forms a hydrophobic barrier to the passage of biologically important molecules. Integral membrane proteins called transporters circumvent the cellular barrier and transport molecules across the cell membrane. These molecular transporters enable the uptake and exit of molecules for cell growth and homeostasis. One important collection of related transporters is the major facilitator superfamily (MFS). This large group of proteins harbors passive and secondary active transporters. The transporters of the MFS consist of uniporters, symporters, and antiporters, which share similarities in structures, predicted mechanism of transport, and highly conserved amino acid sequence motifs. In particular, the antiporter motif, called motif C, is found primarily in antiporters of the MFS. The antiporter motif’s molecular elements mediate conformational changes and other molecular physiological roles during substrate transport across the membrane. This review article traces the history of the antiporter motif. It summarizes the physiological evidence reported that supports these biological roles.

## 1. Introduction 

Bacterial physiology requires the availability of macromolecules and ions, as well as their precise balance concerning the external environment. The cell wall peptidoglycan provides the necessary stability to the cellular structure. In contrast, the cell membrane and its constituent proteins are critical in transporting solutes in and out of the cell in a coordinated manner. Although the transport process involves handling solutes as a major function, the implications of this function are more than the mere movements of substrates, as these processes are necessary for various other activities of bacteria involving metabolism, colonization, communication, virulence, and community living [1,2]. Transporter proteins are a large group of proteins that play critical roles in the physiology of bacteria by transporting essential macromolecules into the cell and extruding toxic metabolites, chemicals, and xenobiotics, maintaining cell homeostasis and helping bacteria survive in a wide range of environmental conditions. These proteins are embedded in the outer membrane of bacteria. They are a transportation conduit and an important means of communication with the external environment. 

Nearly 5–10% of prokaryotic genomes encode membrane proteins, most of which transport water-soluble substrates for cell growth and mediate the expulsion of growth-inhibitory molecules [3]. The transporter proteins differ widely regarding their energization mechanisms, substrate specificity, and structure. For example, passive transporters facilitate the simple diffusion of substances, such as simple carbohydrates and amino acids, down a substrate gradient (downhill). 

In contrast, active transporters transport the substrates against the concentration gradient by an energy-driven process [3,4]. The active transportation process creates an ionic gradient, resulting in a potential difference across the membrane (electro-motive force), which energizes several other cellular operations across the membrane. The path-breaking discovery that homologous transporter proteins of common evolutionary origin transport sugars in prokaryotes and eukaryotes led to the discovery of numerous homologous proteins across domains and the study of their structure–function relationships [5,6]. These proteins were subsequently grouped into several categories based on the substrate profiles, predicted topology, and energy coupling mechanisms [7,8,9]. Whole-genome sequencing projects have enabled the identification and functional elucidation of novel transporter proteins. The advances in in silico predictive modeling have immensely boosted the efforts toward understanding the structure–function relationships of transporter proteins and the identification of novel substrates as well as inhibitors for potential applications in human therapeutics. 

## 2. Transporter Biology 

The transport proteins in living cells employ several different mechanisms to perform the activity and vary widely concerning their structures, types, and range of substrates, as well as the sources of energy that drive the transport process across the biological membrane [8,10]. The bacterial outer cell membrane functions as a protective barrier that selectively allows the movement of solutes across it into the periplasmic area. Since most solutes cannot cross the membrane barrier, specific transporters move substrates into and out of the bacterial cell. The simplest type of solute movement across the cell membrane occurs by passive diffusion of molecules such as certain gases (CO_2_ and O_2_) and water from a higher concentration to a lower concentration (downhill) without the involvement of transporter proteins. On the other hand, facilitated diffusion is enabled by carrier proteins that bind solutes and move them across the membrane through conformational changes. In contrast, channel proteins facilitate the movement of specific molecules through open pores formed by them [11]. As in passive diffusion, facilitated diffusion is not energy-coupled, although the concentration and the electrochemical gradient determine the direction of the movement of the substrate, and is always downhill (Figure 1).

Unlike passive and facilitated diffusion, active transporters, by their ability to transport substrates against concentration (uphill), create solute gradients across the membrane, and this activity is coupled with diverse cellular energy sources. Two types of active transporters are based on the energy sources that drive the transport process. First, primary active transporters, known as ATP-Binding Cassette (ABC) transporter proteins, bind and hydrolyze ATP to drive the active transport of diverse substrates, mostly hydrophilic, such as sugars, amino acids, peptides, lipids, ions, xenobiotics, and drugs into or out of the cell, and also play important roles in the virulence of many pathogenic bacteria [12,13]. Examples of ABC transporters include the vitamin B12 transporter, BtuCD, and the maltose transporter (MalFGK2) from *E. coli* [14,15], the molybdate/tungstate transporter, ModBC from *Archaeoglobus fulgidus*, and the zinc transporter, ZnuABC, of *Bacillus subtilis* [16]. On the other hand, efflux pumps of ABC-type transport drugs and toxic substances out of the cell. Examples include the multidrug transporter Sav1866 from *S. aureus*, BmrA of *B. subtilis,* LmrP of *Lactococcus lactis,* and MacB from *Acinetobacter baumannii* [17,18,19].

The ABC transporters that pump ions across the cell membrane create an ionic gradient that the secondary active transporters utilize to energize their uphill transport activities. 

## 3. Superfamilies of Transporters 

Many transport proteins have been identified over the years. These solute transporters have diverse structures and functions. However, significant degrees of sequence identities and homologies are shared. Thus, a need for classifying these proteins akin to the Enzyme Commission (EC) system for enzymes, based on certain characteristics that distinguish them into distinct groups, was realized. This effort led to the creation of the transporter classification (TC) system (http://www.tcdb.org/ accessed on 24 September 2023), a curated database in which transporter proteins are systematically grouped based on specific characteristics, including the mode of transport, energy coupling mechanisms, sequence homology/protein phylogeny, topology, and substrate specificity [20,21,22,23]. The TC system follows the International Union of Biochemistry and Molecular Biology (IUBMB), an approved method of classification and nomenclature for transport proteins. Proteins originating from a common ancestor are homologous, share similar structures and functions, and are grouped into families or subfamilies. Accordingly, the database has over 1800 families of transport proteins grouped under distinct transporter classes, namely channels/pores, electrochemical potential-driven transporters, primary active transporters, group translocators, transmembrane electron carriers, auxiliary transport proteins, and transport protein families of unknown classification [23,24].

The secondary active transport proteins, i.e., symporters and antiporters, are grouped under the Electrochemical Potential-driven Transporters category and are distinct from the uniporters, which move solutes across the membrane down their gradients. The antiporter proteins transport two molecules simultaneously in opposite directions, energized by the proton-motive force gradient of H^+^ or Na^+^ across the plasma membrane, and are grouped under four superfamilies: (i) the major facilitator superfamily (MFS), (ii) the resistance-nodulation-cell division (RND) superfamily, (iii) the drug/metabolite transporter (DMT) superfamily, and (iv) the multidrug/oligosaccharidyl-lipid/polysaccharide (MOP) superfamily [25] (Figure 2). 

The RND transporters are tripartite structures, forming multi-component complexes with an outer membrane channel and a periplasmic adaptor protein [26]. The multidrug and toxin extrusion (MATE) family of antimicrobial efflux pumps, which use both H^+^ and Na^+^ as energy sources, belong to the MOP superfamily [27,28]. Some of the well-characterized drug/Na^+^ antiporters include YdhE of *Escherichia coli*, NorM of *Vibrio parahaemolyticus* [29], NorM and VcmA of *Vibrio cholerae* [29,30], AbeM of *Acinetobacter baumannii* [31], and BexA of *Bacteroides thetaiotaomicron* [32]. The drug/H^+^ antiporters, such as the QacE and AbeS of *Acinetobacter baumannii*; QacC and SepA of *S. aureus*; EmrE, YnfA and MdtJ of *E. coli* and KpnEF of *Klebsiella pneumoniae*, belonging to the small multidrug resistance (SMR) family, are placed under the DMT superfamily [33,34,35]. 

**Figure 2 biology-12-01336-f002:**
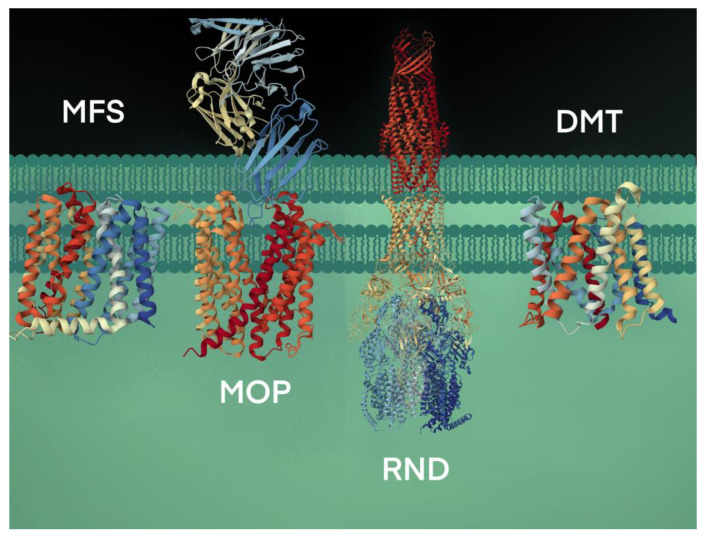
Bacterial efflux pumps of antiport type are grouped under four secondary active transporter superfamilies. The transport of substrates is coupled with H^+^ or Na^+^ ions. The representative MFS protein shown is the crystal structure of MdfA, a multidrug efflux pump (PBD code, 4ZOW) from *E. coli* [36]. The MOP protein shown is a high-resolution crystal structure using cryogenic electron microscopy analysis of NorM, a MATE transporter bound to a Fab molecule (PBD, 7PHP) from *V. cholerae* [37]. The RND transport system shown is the AcrAB-TolC crystal structure (PBD, 5V5S), a multipartite complex from *E. coli* that spans the inner (AcrB) and outer (TolC) membrane and periplasm (AcrA) [38]. The DMT crystal structure is the YddG transporter (PBD, 5I20) from the bacterium *Starkeya novella* [39].

The major facilitator superfamily (MFS) is the largest group of transport proteins widely distributed in Gram-positive and -negative bacteria, which work as secondary active transporters of the symport and antiport type, and passive transport, like uniporters, which undergo facilitated diffusion [40,41]. MFS proteins are typically 400–600 amino acids long and fold into 12–14 transmembrane helices, forming two domains known as N-terminal and C-terminal domains, each composed of six helices and joined by a flexible cytoplasmic loop [42]. The two domains of the MFS pumps are separated by a central cavity involved in substrate binding. Conformation changes induced by substrate binding and the protonation state are engaged in substrate transport using a rocker-switch alternating-access model [36,43,44]. Some of the well-characterized drug/H^+^ MFS efflux pumps are EmrD, YajR, MdfA, and SotB from *E. coli*; QacA [36,43,44,45]; and TetA(K), NorC, and LmrS from *S. aureus* [46,47,48]. Previous cellular and molecular physiological evidence attributes various critical roles to the MFS efflux pumps in antimicrobial resistance, host colonization, toxin secretion, biofilm formation, and cell–cell communication involving quorum sensing in pathogenic bacteria [49,50,51]. 

## 4. Conservation of Amino Acid Sequence Motifs

Early studies of the MFS’s deduced primary amino acid sequences revealed a large degree of shared sequence-relatedness between its members [5]. These reports further demonstrated shared homology between transporters belonging to the MFS [52], suggesting that the MFS proteins have similarities in structure, ancestral origin, and transport mechanisms [53]. Multiple amino acid sequence alignments performed on proteins of the MFS revealed the evolutionary conservation of several signature sequence motifs [54,55,56]. Some of these conserved sequence motifs were shared among most MFS members or functionally related transporter subsets [9]. 

Motif A, characterized by the conserved sequence “Gly-(X)_3_-Asp-Arg/Lys-X-Gly-Arg-Arg/Lys”, is found in the cytoplasmically located loop between transmembrane helices 2 and 3 of virtually all transporters of the MFS [6,54], as shown in Table 1 and Figure 3. The functional importance of this conserved loop structure has been studied extensively in a variety of transporters, such as the LacY lactose symporter of *Escherichia coli* [57], the TetA(B) tetracycline efflux pump from *E. coli* [58], and the LmrP multidrug efflux pump from *Lactococcus lactis* [59]. Residues of motif A have been shown to participate in forming the gating structure, stabilizing the transporter structure, mediating and regulating conformational changes during transport, fashioning the interface between two bundles, sensing ion gradients, and forming a conformational switching system [60]. 

A striking observation in the YajR multidrug efflux pump of *E. coli* showed that sequence elements of motif A were conserved structurally in other loops throughout the transporter, not only in the original loop between helices two and three (L2–3) but in loops between helices five and six (L5–6), between helices eight and nine (L8–9), and between helices 11 and 12 (L11–12) [45]. This observation points to motif A’s functional importance throughout the transporter structure for the canonical MFS protein and potentially all MFS transporters. 

The conserved amino acid sequence of Motif B was denoted as “Arg-(X)_2_-Gln-Gly” and as “Leu-(X)_3_-Arg-(X)_2_-Gln-Gly-(X)_3_-Gly-Gly” and was found in the middle of transmembrane helix 4 of the two largest subfamilies of the MFS [61], as shown in Table 1. Because of the positively charged nature of the Arg residue in the center of a transmembrane helix, it was postulated to play a catalytic role in protonation and energy coupling during symport or antiport [61]. It has been speculated that the arginine residue of motif B in TetA(B) participates in conformational changes that alternately expose the charged residue to the substrate-translocation channel or plays a role in proton transport and energy transduction during transport [61,62,63]. Indeed, after a systematic evaluation of the residues of transmembrane helices 4 and 5 of the TetA(B) tetracycline efflux pump, it was observed that the conserved Arg residue at position 101 lost activity when neutralized [63]. 

Motif D1, denoted as “L d X t v l n v a l p”, where upper- and lowercase lettering represents percent sequence identity greater and lower than 70%, respectively, is found at the C-terminal end of transmembrane one in transporters with 14 helices across the membrane [61]. In comparison, motif D2, “g I g l (X)_2_ P v l P”, resides in the corresponding location, TM-1, in transporters with 12 transmembrane helices [65]. Elements of these motifs (D1 and D2) overlap in the overall consensus sequence identified earlier as “Leu-Pro” in members of both 12- and 14-transmembrane helices [65]. Residues in the first transmembrane domain of MFS sugar-proton symporter LacY were discovered to mediate substrate selection [57]. Hence, it is predicted that residues may play a similar role in the corresponding location of antimicrobial efflux pumps. 

Another conserved sequence called motif E that is found in 14-TM-specific MFS transporters is “Asp-(X)_2_-Gly-(X)_2_-Leu” and resides in the middle of helix seven [61]. Motif F, denoted “l G (X)_3_ G i A v l G X l”, is located in the 13th transmembrane segment of the MFS with 14 helices [61]. On the other hand, motif G, “G P L l g”, appears unique to the MFS transporters with 12 helices and is located in helix 11 [61,65]. Interestingly, residues of motif G seem to be present in the consensus sequence of the highly conserved sequence motif C, also known as the antiporter motif, suggesting an evolutionary duplication and predicting a functional role during transport [9,61,65].

## 5. The Antiporter Motif

Motif C, the MFS antiporter motif, “Gly-(X)_8_-Gly-(X)_3_-Gly-Pro-(X)_2_-Gly-Gly”, was first reported to reside in the center of the fifth helix in transporters with 12 and 14 transmembrane domains in the MFS that are ion-driven antiporters [53,65], as shown in Table 1 and Figure 3. The functional importance of the motif C consensus sequence was first evaluated for the most highly conserved residue, Gly-147, in the tetracycline efflux pump, TetA(C), demonstrating that only serine and alanine were acceptable substitutions [66]. All other amino acid substitutions resulted in a complete loss of tetracycline resistance. In the same study, it was demonstrated, using molecular mechanics analyses, that the fifth transmembrane helix formed a kinked structure [66]. From this systematic mutational analysis of the most highly conserved motif C residue in helix five of TetA(C), it was concluded that elements of the motif confer drug and proton antiport and determine the structural features necessary to mediate changes in the orientation of the substrate binding site [66]. 

A structure–functional analysis of the TetA(K) tetracycline efflux pump from *S. aureus* was conducted in which the remaining conserved residues of motif C were mutated, showing reduced drug resistance in the mutants, thus establishing the importance of these residues for antimicrobial efflux [67]. Further, because of the relative abundance of the glycine residues in motif C, it was proposed that these residues conferred a relatively high degree of molecular flexibility, predicting that the structure formed by the fifth helix plays a role in conformational changes during transport. Likewise, a cysteine-scanning mutagenesis study of the motif C residues in TetA(B) from *E. coli* showed they were largely necessary for tetracycline/proton antiport activity and suggested that helix five formed a permeability barrier to prevent unwanted ion leakage and collapse of the membrane potential [63]. An interesting study of second-site suppression analysis of a defective primary mutation at Gly-247 of TetA(B) showed complementary mutations that restored resistance activity in helix five [68]. Restoring inactive mutations by compensating second-site mutations supported the role of motif C in forming the protective permeability barrier and conformational changes. 

The conserved Gly-Pro dipeptide of the antiporter motif became important when structure–function and molecular modeling of helix five showed a kinked structure in TetA(C) [66]. The physiological role of the Gly-Pro dipeptide was studied in TetA(L) of *Bacillus subtilis*, demonstrating that this structure was necessary for substrate binding and preventing ion leakage [69]. The proline of the Gly-Pro was evaluated in QacA from *S. aureus*, showing decreased resistance levels to antimicrobial agents [70]. The structure formed by the residues of the Gly-Pro dipeptide became the focus of molecular dynamics simulations and homology modeling in the VAChT vesicular acetylcholine transporter from various eukaryotic species, ranging from fungi to humans [71]. Sequence analysis of a pump from *S. aureus*, a 14-membrane spanning protein called Tet38, a tetracycline efflux pump that also transports fatty acids, tunicamycin, and fosfomycin, was further shown to harbor motif C [72]. In a more recent study, the glycine of the Gly-Pro dipeptide in Tet38 was altered by mutagenesis and demonstrated to lose tetracycline efflux activity [73].

As predicted [66], the Gly-Pro residues in helix five form a kinked structure [74]. In the latter study, the kinked membrane-embedded structure involving motif C’s helix was flexible, exhibiting wobbling behavior and lowering the energetic barrier. As mentioned above, in the MdfA structure, the fifth helix harboring motif C was kinked as predicted [66]. Interestingly, the TM5 helix underwent rotational twisting as the substrate translocated through the transporter [75]. The structural dynamics involving these residues of motif C were consistent with the notion that helix five of the transporters in the MFS line the interface region between the molecular bundles formed by helices 1–6 and 7–12 of the VAChT protein, a situation predicted for GlpT [76]. The flexible kinked nature of the structure formed by motif C was studied in the CaMdr1p multidrug efflux pump from *Candida albicans*, a eukaryotic microorganism, showing that these conserved residues mediated tight helical packing [77]. Along these lines, the VAChT vesicular acetylcholine transporter from *Rattus norvegicus* showed the loss of acetylcholine transport after the conserved Gly-Pro dipeptide residues of motif C were altered by mutation [78]. These physiological effects on substrate transport across the membrane indicate that motif C confers a kinked intra-membrane helix and influences a flexible conformational switch while forming a tight seal that prevents undesirable proton leakage [74,78]. In the related vesicular monoamine VMAT2 transporter from *R. norvegicus*, the structure dictated by residues of motif C formed a molecular hinge, characterized by helices five and eight interacting with helices two and eleven converging at the bundle interface during substrate transport [79]. 

Another function dictated by residues of motif C appears to be the determination of transporter substrate specificity. The Mdt(A) multidrug efflux pump (previously denoted as Mef214) encoded on a plasmid isolated from *Lactococcus lactis* showed resistance to several structurally unrelated antimicrobial agents [80]. In the milk pathogen *Lactococcus garvieae*, the Mdt(A) pump showed differences in the substrate resistance profile, such as enhanced susceptibility to the macrolide erythromycin, in a protein variant where a valine at 154 was altered to phenylalanine [81]. In the same study, another motif C-like sequence in helix nine of Mdt(A) showed that Ile-296 changed to Val, in which tetracycline resistance was reduced. Interestingly, Mdt(A) appears to harbor an ATPase domain, indicating a departure from the energetic nature of secondary active transport to that of a primary energetic constitution. Additional study of this transporter is needed to understand the types of driving forces involved in mediating antimicrobial transport across the membrane. This issue of energetics is of importance for other transporters of the MFS. 

MdfA is a well-characterized MFS multidrug efflux pump from *E. coli*, previously known as Cmr and CmlA [82]. Three high-resolution crystal structures were solved for the MdfA transporter: one bound to the substrate chloramphenicol, the second structure attached to deoxycholate, and the third bound to *n*-dodecyl-*N*, *N*-dimethylamine-*N*-oxide, the latter two of which are known substrate analogs [83]. Strikingly, all three substrate-bound structures showed the characteristic kinked transmembrane helix five closely associated with the regions with bound ligands. In MdfA, the TM5 helix represents one of a set of so-called rocker helices, described as participating in a series of pseudo-symmetrical helical repeat structures involving three helices, a structural feature observed four times in symporters of the MFS, such as GlpT [84]. Another striking observation was that the chloramphenicol binding site in MdfA at helix one was surrounded and perhaps protected by residues in the motif C helix, such as Val-149, Ala-150, Ala-153, and Pro-154, forming the critical interface structure between the two bundle domains typical of the MFS transporters. Thus, it is established that TM-5 of the MFS efflux pumps constitutes part of the interface between the N- and C-terminal domains. This bundle-interface role for conserved residues of the fifth membrane helix is consistent with the notion that motif C stabilizes the pump’s bundle structure to maintain a cytoplasmic-facing configuration and prevents ion (proton or sodium) gradient dissipation, which would otherwise deplete the energetic driving force of secondary active transport [83]. 

In another study, two crystal structure versions of the MdfA molecule were isolated [85]. One of these showed the MdfA pump bound to the substrate acetylcholine. In contrast, the other MdfA structure was attached to the efflux pump inhibitor reserpine. Whereas the two MdfA ligand binding sites are distinct, the fifth helix containing residues of motif C is involved and plays an important role in each situation. In the case of the MdfA structure bound to acetylcholine, TM5 appears to form a conformation that provides a protective function or helps to stabilize the substrate binding pocket for acetylcholine.

On the other hand, motif C seems to play a more direct role in the MdfA structure bound to reserpine [85]. The proline residue at position 154, a component of the Gly-Pro dipeptide of motif C, interacts closely with reserpine. The MdfA structure bound to reserpine appears to prevent the availability of Asp-34 for the transport of acetylcholine, shedding light on the molecular mechanism of efflux pump inhibition. 

According to studies on protein crystal and X-ray diffraction, MdfA is a 12-TM transporter [86]. Conversely, the nature of the fifth helix in MFS transporters with 14 TMs remains unclear. In a recent report, however, it was shown that in a cryo-EM structure of the *S. aureus* QacA efflux pump, which has 14 TMs, the fifth transmembrane helix was kinked [86]. Previously, it was established that when Gly-143 of QacA was changed to glutamate, Pro-144-Arg, or Gly-147-Asp, all residues that are conserved in motif C, and antimicrobial susceptibilities were enhanced, supporting the functional role of the kinked helix [83]. A recent structure–function study of motif C residues in helix five of the novel sugar efflux pump, SotB from *E. coli*, showed that Gly-153 and Gly-157 were necessary for arabinose transport [44]. It was postulated from these SotB mutants that motif C functions in forming the substrate binding pocket and linking the transport of protons during sugar efflux. 

It should be noted that with more refined primary sequence alignment software [87], critical elements of the motif C sequence, namely the glycines, also appear in symporters [79], indicating that it is relevant to the physiological operation of most of the MFS transporters. The lactose-H^+^ symporter, LacY, of *E. coli* lacks the Gly-Pro dipeptide [57]. The corresponding residues of the motif C in TM5 of LacY are “Gly-(X)_8_-Gly-(X)_3_-Cys-Ala-(X)_3_-Gly”, with the “Cys-Ala” sequence in place of the Gly-Pro dipeptide that is observed in the antimicrobial antiporters of the MFS [88]. In a systematic evaluation of helix five in the LacY transporter using cysteine-scanning mutagenesis, the alteration of several residues corresponding to motif C showed a significant reduction in lactose transport activities across the membrane [89]. Prominent among these affected residues in the lactose symporter is Gly-147, equivalent to the second glycine of the canonical motif C sequence, and Cys-154, the third glycine of the consensus sequence of motif C, which corresponds to the glycine of the Gly-Pro dipeptide [57]. Additional residues of the motif C that are reduced in transporting lactose when replaced with cysteine include Gly-150 and Gly-159, the second and last glycine of the motif C sequence, respectively [57,88,89].

Interestingly, two glycines adjacent to each other in the motif C sequence flank an Arg-144 residue, forming a salt bridge with Glu-126 [90], presumably stabilizing the local structure around the critical charge-charge pair. It is striking that Arg-144 of LacY is not conserved in other symporters, like the GlpT glycerol-3-phosphate transporter, which is also a member of the MFS and has an Asn in place of the Arg [84]. Likewise, the intra-membranous arginines are lacking in many of the antiporters of the MFS [79], suggesting that this part of the energy transducing system that utilizes cation gradients across the membrane is distinct between symporters and antiporters. 

## 6. Conclusions and Future Studies

Because the vast array of the MFS integral membrane proteins is harbored across all known living taxa, these solute and ion transporters are functionally critical to the biology and chemistry of the biological cell in all such organisms. Analyses of the deduced primary sequences of the MFS transporters in light of the structural studies emerging from various laboratories have provided critical molecular insight regarding the nature of solute transport across the biological membrane [91]. In particular, several highly conserved amino acid sequence motifs have become crucial to understanding the translocation process as substrates and ions pass across the membrane through the homologous and related MFS proteins. Among these shared sequence motifs is the critical antiporter motif C, a string of highly conserved amino acids that occupy a key transmembrane helix and are shown to play various functional roles while transporting a tremendous array of structurally different biologically important substrates. 

There are numerous molecular biological roles played by residues and molecular structures dictated by the motif C. Prominent among these roles involve helping to form a central binding cavity for substrates, modulators, and efflux pump inhibitors; influencing the direction of solute transport across the membrane; mediating conformational changes of the MFS transporters during substrate and ion translocation through the channels; and the maintenance of the energy gradients to prevent undesirable ion leakage that would collapse the membrane potential. Further, motif C residues can serve structural roles, such as influencing protein structure stability, protecting transporter-bound substrates, providing a contact interface between the two large transport bundles, forming a molecular hinge structure that twists about during substrate and ion transport, and comprising a conformational switch system and a regulator of that transporter conformation switching. It is anticipated that future studies in which conformational states of motif C at the structural level are combined with efflux pump inhibitor complexed with MFS transporter will hold tremendous promise but are currently lacking investigation. 

Originally thought to have been a feature critical solely to the MFS antiporters, motif C’s residues can also be detected in MFS symporters. Therefore, its presence in MFS symporters gives the amino acid residues of the membrane helix five enhanced and universal importance in transporters of living cells from virtually all organisms. Motif C in antimicrobial antiporters of the MFS can serve as suitable molecular targets for modulation of antimicrobial and anti-cancer resistance to improve and restore clinical outcomes of infectious disease and cancer chemotherapies. The MFS’s antimicrobial, multidrug, and anti-cancer drug efflux pump systems are known to harbor motif C’s highly conserved signature sequence [49,50,51]. In this new light, we propose that the structures produced by motif C be considered a virulence factor harbored by microbial pathogens and cancer cells [60,64]. 

Likewise, motif C residues and their structures can potentially be exploited to engineer host cells to acquire industrially and biotechnologically desirable substrates or perhaps to secrete them from cells modified to manufacture them intracellularly. Thus, transporters of the MFS and the conserved sequence motifs they share, especially motif C, represent an untapped resource for enhancing productivity in biomedical and basic biological research programs. 

Despite these past and recent advances in our understanding of the various biological roles of the molecular structure formed by motif C residues throughout the years and the vast potential for advancement in basic and applied research, several matters remain unclear. For instance, it is poorly understood how motif C is involved in mediating non-specific ion leakage prevention while simultaneously permitting the transport of larger substrate and ions necessary for symport and antiport. Along these lines and as of this writing, it remains unclear whether motif C influences the energy-transducing system in a general, universal way for the majority of the MFS transporters or if the bioenergetics role played by motif C is unique to each transporter, as the functional roles of the non-conserved residues that are interspersed throughout the TM5 are poorly studied. The role of conserved motif C residues versus non-conserved counterparts within helix five of transporters in the MFS in transferring the energy stored in secondary active systems into substrate translocation through a transporter is uncertain. 

Regarding the relationship between motif C and energy-transduction during transport, the relationship between the ATPase system and the secondary active transport mechanism definitively known to operate in MFS proteins is particularly interesting, especially as MFS transporters, like LmrP, continue to be studied in detail [92,93]. On a related note, the relationship between motif C residues and the mechanism of solute transport in passive transporters of the MFS proteins is a neglected field of investigation. Indeed, much research is needed to gain a molecular and cellular understanding of motif C’s role in passive versus active transporter systems. Along these lines, it remains to be understood how transporters determine whether and how to invoke passive and active solute transport in the cell. 

As new studies regarding efflux pump inhibitors for MFS transporters are numerous and regularly reported, this area appears to suffer from a lack of translation to clinical treatment against infectious diseases and cancers. It is clear that clinical trials are sorely lacking and needed, as new cases of multidrug-resistant infection and cancer are currently alarming and projected to increase in the coming years significantly. New strategies are required to adequately address the problem of drug resistance, whether in microorganisms or cancer, and studies of the biology of residues in motif C hold promise [94]. Our laboratories have studied the effects of various modulators of MFS efflux pumps, such as ErmD-3 from *V. cholerae* and LmrS from *S. aureus*, both transporters of which are bacterial multidrug efflux pumps of the MFS [48,95,96]. Our future work will focus on the molecular and cellular roles in these modulatory effects and motif C. We anticipate that investigators who are novel MFS efflux pump inhibitor designers shall consider the helix five with motif C a suitable target for designing such modulators. 

The binding of various substrates to residues in helix five of the MFS transporters poses a question on the extent to which motif C is involved in mediating substrate specificity. It is unclear whether motif C influences substrate specificity directly or through indirect means by remotely stabilizing binding pockets. Further studies on the relationship between ion- and substrate-specificities are dictated by the functional roles known to be mediated by motif C. For instance, it is desirable to understand the molecular structural and biochemical natures of motif C more clearly during each stage of the symport versus the antiport processes in MFS transporters. Presently, only parts of these transport mechanisms are understood. The full molecular mechanistic story is needed to understand the biology of solute transport as conferred by motif C in proteins of the MFS. 

As we consider the progress made thus far regarding the various functional roles conferred by motif C, we acknowledge that much work remains to be investigated on many fronts, ranging from basic to applied and industrial research. These fronts involve the mechanisms of solute and ion translocation across the membrane, conformational changes in transporters during transport, the energetic driving forces involved during transport, the molecular mechanism of substrate specificities, and our efforts to modulate these various transport systems. We anticipate that analyses of the different molecular structures inherent in the MFS transport proteins and their relation to highly conserved amino acid signature sequences will shed new light on the physiological and molecular biological mechanisms that drive the operation of these transporters in all known living cells and organisms. Much work in studying the disparate members of the MFS remains to be performed. 

## Figures and Tables

**Figure 1 biology-12-01336-f001:**
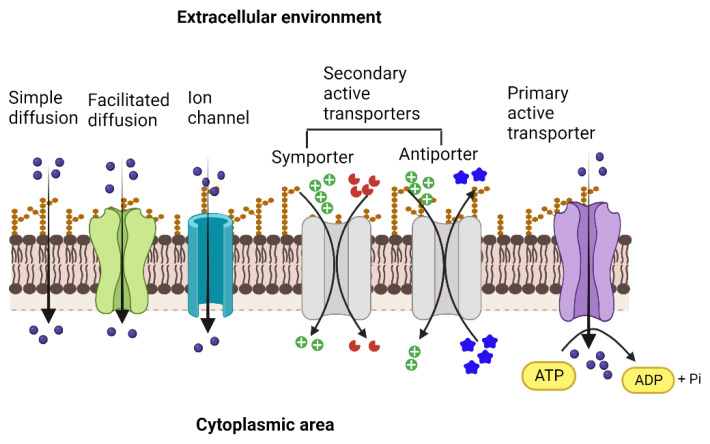
Types of membrane transporters that enable the movement of ions and solutes across the membrane into the cytoplasmic area and vice versa. Simple diffusion of gases and ions across the membrane facilitated diffusion and movement through ion channels (ungated or gated) that occur down the concentration gradient and are not coupled with energy sources. Both primary active and secondary active transporters move the solutes against the concentration gradient. They are energized either by the hydrolysis of ATP (primary active) or by the movement of ions, such as protons or sodium, driven by the electrochemical gradient across the membrane (secondary active). In the case of secondary active transport, the energetic driving force of one solute moving down its electrochemical gradient is coupled to the movement of the other solute moving up its concentration gradient. Created with BioRender.com.

**Figure 3 biology-12-01336-f003:**
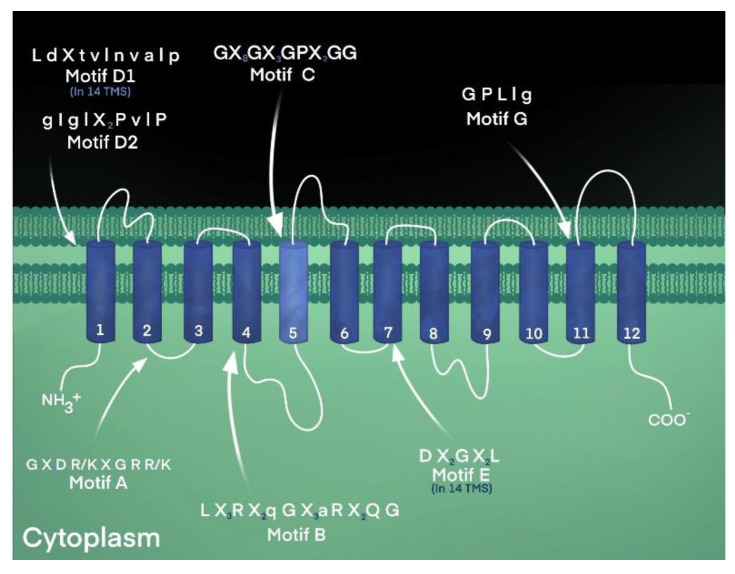
Conserved amino acid sequence motifs of the MFS. Motif C, the antiporter motif, in an MFS transporter’s fifth membrane-spanning domain. The highly conserved signature sequence is found in MFS proteins that confer ion-substrate antiport, including proteins with 12- and 14-TM segments. The antiporter motif is characterized by a high glycine content and a highly conserved proline residue, a known helix-breaker. Other sequence motifs are indicated in the locations reported [9].

**Table 1 biology-12-01336-t001:** Amino acid sequence motifs in transporters of the MFS.

Motifs Consensus Sequences *	Locations	Functions	References
**Motif A** G X_3_ D R/K X G R R/K	Loop between TM2 and TM3 (Loop2–3)	Gating, transporter stability, conformation change regulator, bundle interface stability, ion gradient sensor, conformational switching	[6,53,60]
**Motif B** L X_3_ R X_2_ q G X_3_ a R X_2_ Q G	TM4 of sub-family 3	Proton binding and transport, energization, conformational changes	[61,62,63]
**Motif C** G X_8_ G X_3_ G P X_2_ G G	TM5 of antiporters	Antiport, conformational changes, permeability barrier, ion leakage prevention, substrate binding and specificity, molecular hinge, bundle interface forming, substrate-binding pocket stability, target for efflux pump inhibition, conformational switch regulator	[60,64,65,66]
**Motif D1** L d X t v l n v a l p	C-terminal end of TM1 in MFS proteins with 14-TMS	Unknown, predicted to bind substrate and mediate substrate selection	[9]
**Motif D2** g I g l X_2_ P v l P	C-terminal end of TM1 in proteins with 12-TMS	Unknown, but postulated in substrate binding and specificity	[9]
**Motif E** D X_2_ G X_2_ L	TM7 of 14-TMS MFS proteins	Conserved Asp is predicted to bind and transport protons	[61]
**Motif F** l G (X)_3_ G i A v l G X l	TM13 of 14-TMS proteins	Unknown	[9]
**Motif G** G P L l g	TM11 of 12-TMS proteins	Evolutionary duplication of motif C, and thus postulated to play similar functional roles during transport	[9,61]

* Reported consensus sequences are provided. Upper- and lowercase letters represent the one-letter amino acid residue codes reported to be greater and lower than 70% sequence identity, respectively. X represents any amino acid. In some cases, more than one consensus sequence is presented, as reported in the literature.

## Data Availability

Not applicable.

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
