# Peer review of "Functional Roles of the Conserved Amino Acid Sequence Motif C, the Antiporter Motif, in Membrane Transporters of the Major Facilitator Superfamily"

_biology, 2023, doi:10.3390/biology12101336_

Round 1

Reviewer 1 Report

This manuscript is a well-written review of antiporters in the MFS superfamily of transporters, with a particular focus on motif C. I think the community could benefit from the publication of such a review. However, there are some minor issues that I think should be addressed, mostly regarding the precision of language in same cases, and also some recommendations for figures.

Figure 1 could benefit from labeling the symporter and the antiporter. Also, rather than showing one solute moving up its gradient, it would be more precise to show a second, coupled solute moving down its gradient. Lastly, the figure caption states that secondary active “are energized either by the hydrolysis of ATP (primary active) or by the movement of protons driven by the electrochemical gradient across the membrane (secondary active).” While secondary active transporters *can* be driven by protons, they need not be. They can also be driven by a sodium gradient, for example (as in sodium-calcium NCX exchangers). More precise language would indicate that the energetic source driving secondary active transporters would be one type of solute moving down its electrochemical gradient, coupled with another solute going against its gradient.

Lines 119-123: Please break up this very long sentence so that it is more readable.

Line 136 refers to uniporters as potential secondary active transporters. This cannot be true because secondary active transporters, by definition, must couple the favorable movement of one solute down its gradient with the movement of another solute against its gradient. A uniporter secondary active transporter therefore cannot exist.

Figure 2 needs a more expansive caption. Beyond being representatives of the 4 transporter classes indicated, what specific proteins are these? Are these experimental structures? If yes, say so, including their PDB codes. If they’re AlphaFold models, say so, including for which proteins they are models. Either way, this information should be in the caption.

Lines 167-169 once again indicate that MFS transporters can be uniporters, symporters, or antiporters (true), but also indicates that all of these are secondary active transporters, which is not true. GLUT1, for example, is an MFS transporter that is a uniporter supporting facilitated diffusion of glucose, but such facilitated transporters are not and cannot be secondary active transporters.

In Fig. 3, the motif is labeled GX8GX3GPX2GG, while in line 238 it is indicated GX3GX3GPX2GG. Please resolve (specifically the first numeral being an 8 vs a 3).

Figure 3 feels incomplete/sparse. I would recommend one of two things. Either use some of the ample space remaining to label the other motifs discussed in the prior section. Or alternatively, display the actual structure of an antiporter and label motif C in the structure. As it stands, using the cartoon schematic with only one motif labeled seems like unnecessary use of space for a schematic.

Author Response

Reviewer 1: Comments and Suggestions for Authors

This manuscript is a well-written review of antiporters in the MFS superfamily of transporters, with a particular focus on motif C. I think the community could benefit from the publication of such a review. However, there are some minor issues that I think should be addressed, mostly regarding the precision of language in same cases, and also some recommendations for figures.

  1. Figure 1 could benefit from labeling the symporter and the antiporter. Also, rather than showing one solute moving up its gradient, it would be more precise to show a second, coupled solute moving down its gradient. Lastly, the figure caption states that secondary active “are energized either by the hydrolysis of ATP (primary active) or by the movement of protons driven by the electrochemical gradient across the membrane (secondary active).” While secondary active transporters *can* be driven by protons, they need not be. They can also be driven by a sodium gradient, for example (as in sodium-calcium NCX exchangers). More precise language would indicate that the energetic source driving secondary active transporters would be one type of solute moving down its electrochemical gradient, coupled with another solute going against its gradient.

Authors’ reply: We thank the reviewer for this and other comments. We fixed Figure 1 and clarified the caption to indicate the ion, i.e., proton versus sodium gradients, and the solute movements as driven by the electrochemical gradients of one solute driving the transport of the other against its gradient in the caption.

  1. Lines 119-123: Please break up this very long sentence so that it is more readable.

Authors’ reply: We revised the sentence by breaking it into three.

  1. Line 136 refers to uniporters as potential secondary active transporters. This cannot be true because secondary active transporters, by definition, must couple the favorable movement of one solute down its gradient with the movement of another solute against its gradient. A uniporter secondary active transporter therefore cannot exist.

Authors’ reply: This is also an excellent point. We fixed the sentence to separate the uniporters from the secondary active transporters, hopefully making the appropriate distinction between the two transporter types.

  1. Figure 2 needs a more expansive caption. Beyond being representatives of the 4 transporter classes indicated, what specific proteins are these? Are these experimental structures? If yes, say so, including their PDB codes. If they’re AlphaFold models, say so, including for which proteins they are models. Either way, this information should be in the caption.

Authors’ reply: We included the requested PDB code information and names of the transporters in the caption of Figure 2.

  1. Lines 167-169 once again indicate that MFS transporters can be uniporters, symporters, or antiporters (true), but also indicates that all of these are secondary active transporters, which is not true. GLUT1, for example, is an MFS transporter that is a uniporter supporting facilitated diffusion of glucose, but such facilitated transporters are not and cannot be secondary active transporters.

Authors’ reply: We corrected the passage regarding symport and antiport, distinguishing them from passive transport, such as uniport.

  1. In Fig. 3, the motif is labeled GX8GX3GPX2GG, while in line 238 it is indicated GX3GX3GPX2 Please resolve (specifically the first numeral being an 8 vs a 3).

Authors’ reply: We appreciate this comment. The first X should be 8, and we corrected it accordingly.

  1. Figure 3 feels incomplete/sparse. I would recommend one of two things. Either use some of the ample space remaining to label the other motifs discussed in the prior section. Or alternatively, display the actual structure of an antiporter and label motif C in the structure. As it stands, using the cartoon schematic with only one motif labeled seems like unnecessary use of space for a schematic.

Authors’ reply: We revised Figure 3 to include the other motifs. We thank the reviewer for each of their comments.

Reviewer 2 Report

 Suggestions

1. A table listing the sequences, locations and functions of all motifs found in MFS, i.e., motifs A, B, C, Ds, E, F, and G should be considered.

2.   ln 413-414. Authors indicated that motif C is also present in the MFS symporters (reference 82). However, the reviewer could not find this information in the cited reference. According to reference 82, the motif on TM5 of rVMAT2 is “ X2-G-X3-G-X3-G-X3”, which missing the important GP motif of “motif C”. This point should be clarified.

Minor points:

1.   ln 141: ((?

2.   Fig. 2 legend: information on the composition of each pump should be included.

3.  ln 188: ((?

4.  ln 257-258: unclear. Mutations in the remaining conserved residues showed reduced resistance?

5.  ln 308: Spell out “L.”

Author Response

Reviewer 2: Comments and Suggestions for Authors

 Suggestions

  1. A table listing the sequences, locations and functions of all motifs found in MFS, i.e., motifs A, B, C, Ds, E, F, and G should be considered.

Authors’ reply: We thank the reviewer for this and the other comments, all of which were helpful. We included the Table with motif sequences, locations, and functions.

  1. ln 413-414. Authors indicated that motif C is also present in the MFS symporters (reference 82). However, the reviewer could not find this information in the cited reference. According to reference 82, the motif on TM5 of rVMAT2 is “ X2-G-X3-G-X3-G-X3”, which missing the important GP motif of “motif C”. This point should be clarified.

Authors’ reply: rVMAT2 is an antiporter with the GP dipeptide, as shown in Figures 1C and S1 of the reference we cited (now 90). According to that reference, Yaffe et al. 10.1073/pnas.1220497110, in Fig. 1C, rVMAT2 has “X2-G-X3-G-P-X2-G,” and in Fig. S1 of the same reference, rVMAT2 has “G-G3-G-X3-G-P-X2-G.” The canonical motif C sequence is G-X8-G-X3-G-P-X2-G-G. Yaffe et al. had further observed that TM5 of LacY is rich in glycines. Perhaps this reviewer meant that LacY is missing the GP dipeptide, an observation we had alluded to when we originally said that LacY has “Cys-Ala” instead of “Gly-Pro.” In any case, we further revised the passage to clarify that LacY, a symporter, is missing the GP dipeptide. Furthermore, we clarified the passage to indicate that in LacY of the MFS, its TM5 is rich in glycines (these were the “critical elements” we had referred to in our original submission).

Minor points:

  1. ln 141: ((?

Authors’ reply: The extraneous parathesis was removed.

  1. 2 legend: information on the composition of each pump should be included.

Authors’ reply: Yes. Reviewer one had a similar issue. We included in the figure caption the transporter names, references, microorganisms, and PDB codes.

  1. ln 188: ((?

Authors’ reply: The extra bracket was deleted.

  1. ln 257-258: unclear. Mutations in the remaining conserved residues showed reduced resistance?

Authors reply: We corrected the sentence to indicate that mutations of the other motif residues showed reduced resistance.

  1. ln 308: Spell out “L.”

Authors’ reply: We spelled out Lactococcus. We thank this reviewer for their commentary.